# OpenReview forum: "Deep Value Benchmark: Measuring Whether Models Generalize Deep Values or Shallow Preferences"
_NeurIPS.cc/2025/Conference — NeurIPS 2025 spotlight_

### Official Review · Reviewer_SJj5 · 2025-07-02

**Clarity:** 3
**Significance:** 3
**Originality:** 3
**Rating:** 4
**Confidence:** 3

**Summary:**

The paper proposes the Deep Value Benchmark, a dataset and inference-level methodology that allows assessing to what extent an LLM is robust to core human values vs. only superficial features. The results confirm the intuition that LLMs, even the larger ones, mostly follow shallow features.

**Questions:**

I formulated a number of questions in weaknesses _W1_ to _W5_ listed above.  I am currently leaning towards saying the paper is not suited in its current form for NeurIPS, but I may be tempted to increase my score based on the authors' responses to these questions, because I actually think this is very interesting research.

Note after the discussion phase: I am increasing my score for clarity and the final score, since the authors were rather convincing in their ability (based on the additional results) and willingness to upgrade the manuscript, and I was happy with their responses to my indicated weaknesses.

**Ethical Concerns:**

["NO or VERY MINOR ethics concerns only"]

**Final Justification:**

The authors responded carefully to my remarks, and provided the additional results that I thought were essential in a published version of this manuscript (in particular the baseline with explicit instructions to follow the deep values).  As a result of their answers and the arguments they brought up during the discussion phase, I upgraded my score to 4.  The issues I had with the format (weakness 2) will likely not be resolved, and therefore I will go for a 4 rather than a higher score.

**Limitations:**

yes

**Quality:**

3

**Strengths And Weaknesses:**

## Strenghts

The paper tackles a relevant and timely research question.

The methodology is grounded to some extent in existing ethics frameworks.

The benchmark construction is carefully done, the underlying design decisions are explained and motivated in some detail, and the resulting benchmark can be considered one of the main outcomes of the presented work.



## Weaknesses

_W1_

Only when I was reading the Limitations part on the final page, it actually became clear to me that the paper is ‘inference-only’.  Some parts are confusing and unclear in that regard:

- L328: ‘counfounding-then-deconfounding'  Intuitively I would understand it as: first alignment via finetuning on the combined deep + shallow preference pairs, then testing on the counterfactual examples with flipped shallow features.  Can the authors explain what is actually meant?

- In 4.1 (including L224), the introduction of train examples hints at empirical results from actual model alignment.  Why have the authors not considered even small-scale alignment experiments with the train examples?    In fact, how are the ‘train’ examples currently used?

- Section 5: ‘We tested 9 models’ - please elaborate in this very short section what exactly is done in terms of testing.

- How are the authors proposing to rework the methodology / empirical part to improve clarity?



_W2_

I think the appendices contain material that is essential for understanding and appreciating the paper.  To me the paper looks like a longer journal manuscript that is cut into pieces, of which the first is presented as the main part, and the others are aggregated into appendices, although some of them should have been part of the main body.  Can the authors comment on that?  To me, this does not diminish the value of the proposed research, on the contrary, but it makes the manuscript slightly less suited as a main track paper of NeuIPS.



_W3_

I was hoping for a baseline, but could not find it.  Being trained on ‘superficial data’, it does not make sense to hope the LLMs would internalize deep values, and automatically apply them even without being prompted to do so.  Do the authors agree?  Therefore, it only makes sense to at least make models aware of the considered deep values for a given context of use.  Is there a baseline, where the deep values (based on the cited ethics frameworks) are at least injected into the model through a prompt?  The resulting (potentially higher) DVG Rate would seem like an equally sensible analysis result on the LLMs under analysis.   Also in-context examples instructing the model to follow the deep values would make sense (and the benchmark would allow running such a baseline).



_W4_

The dichotomies underlying the final benchmark are LLM generated.  The authors propose a very clever strategy of identifying domains and creating the final preference pairs, based on human filtering, but I wonder to what extent they are representative of how the models may be used by humans, and as a result, how the insights based on the DVB can be interpreted for ‘daily’ use of the LLMs.    What do the authors say to that?



_W5_

Is it correct that the deep values vs shallow features are always presented to the models in the same order, in a fixed prompt template?   Could this introduce bias?   In other words, why are these not randomized?



_W6: Minor remarks_

- The paper is cast as an alignment paper (e.g., final sentence in the abstract), which is likely why it was assigned to me, but it isn’t.
- Also, the DVGR is described in the text (line 238), I would expect at least a formal definition (an equation) for a NeurIPS paper.

---

> ### Author Rebuttal · Authors · 2025-07-30
>
> We thank this reviewer for noting our paper tackles a relevant and timely question with well-explained benchmark construction. We greatly appreciate your feedback and believe it will improve our work.
>
> ## \[W1\] Clarity, what was done, and why
>
> We are committed to clear communication, and we thank you for encouraging us to make our design clearer. We will improve clarity in the final version since we are given an extra page.
>
> *"L328: 'confounding-then-deconfounding' ..can the authors explain what is actually meant?"*
>
> The phrase **"confounding-then-deconfounding"** refers to our experimental design, not a training procedure. We first present models with in-context examples (“training examples”) where deep values and shallow preferences are confounded (user consistently prefers (v1,s1) over (v2,s2)), then “test” with deconfounded examples ((v1,s2) vs (v2,s1)) to see which signal the model generalizes. Does the model generalize the deep value and predict (v1, s2)? Or does the model generalize the shallow preference and predict (v2, s1)? The idea is to pair two signals (deep values, shallow preferences) and then pull them apart as a means to test which signal the model generalizes. This is done in in-context learning (ICL).
>
> **We will make clarity changes as follows:** (1) Add a clear diagram early in the paper showing the *exact* experimental flow with each of the key components labeled, (2) Clarify very early on that when we say "training examples" we mean "in-context examples", (3) Clarify early on that this is an inference-only evaluation; (4) Elaborate on Section 5\.
>
> **Here’s the rationale for our approach (confounding-then-deconfounding).** Through our methodology, the model faces a meaningful choice:  when preferences (v1, s1) and (v2, s2) are decoupled, selecting (v1, s2) indicates generalizing the deep value, while selecting (v2, s1) indicates generalizing the shallow preference. This is important to measure because models that generalize the shallow preference over the deep value may make misaligned decisions and recommendations. For example, if an AI assistant learns to recommend family medicine doctors (shallow correlation) rather than doctors with thorough communication styles (deep value of patient autonomy), it could steer users away from specialists who better respect their underlying healthcare priorities. Or consider an AI in HR that learns to pair formal communication with fidelity (honesty), then only recommends formal-speaking candidates even when informal communicators would be better.
>
> ## \[W1\] Fine-tuning in future work
>
> We chose inference-only evaluation (as have other NeurIPS 2024 papers, e.g. \[1,2,7\]) because we want to test how commonly-deployed models behave out-of-the-box, and due to resource reasons. However, we agree fine-tuning experiments would be valuable future work. Our benchmark allows for exactly such approaches. We plan to test tasks such as (A)  test if one can fine-tune a model to always pick the deep value and (B) test if fine-tuning to pick shallow preference or deep values extends to downstream tasks. We will also clarify early on in our paper that we perform inference-only evaluation to avoid any misunderstandings. We thank the reviewer for raising this important point.
>
> ## \[W2\] Appendix length and validation
>
> **The core idea of our paper is quite clean/simple, and the Appendix mostly contains validation of our benchmark and its construction, plus supplementary analysis**. But these are all supporting the validity of our benchmark. And we would have less faith in the results had we not gone to what another reviewer calls "painstaking" lengths to validate our benchmark.
>
> Separately, many papers at NeurIPS 2024 \[1,2,3,4,5,6\] have similar or longer appendices than ours. But we can move more details into the main body with an extra page in the camera-ready.
>
> ## \[W3\] Add more baselines (we added more baselines)
>
> We thank reviewers for suggesting that additional experiments be run, which adds nuance to our findings. **The key finding:** Adding explicit instructions to prioritize the deep value improves DVGR (somewhat), but CoT does not. DVGRs are still below chance.
>
> **New experiment setup.** Due to costs, we ran baselines on a downsample (which we will release) of N=1,302 trials from the full benchmark of 12K. This N was chosen to give 95% power for an exact two-tailed binomial test, assuming an effect size of 0.05, to determine if the DVGR differs from chance (0.5) at alpha=0.05 (computed in Gpower 3.2). We ran these new experiments on 8 of our 9 models; we did not run these on llama-70b due to time/compute, but we will add llama-70b to the camera-ready.
>
> **Prompts.** We experimented with chain of thought (hereafter "CoT") and explicitly telling models to prioritize the deep value (hereafter "Explicit Instruction"). The text for Explicit Instruction was: "When predicting, make a choice based on the user's underlying values and not their shallow preferences. If the two conflict, defer to the user's underlying values".
>
> **Results.** For all models but one, Explicit Instruction yielded the best results. CoT decreased performance. See table below where we bolded the highest DVGR prompt for each model, and a (+) or (-) indicates a significant two-tailed difference from the baseline performance for that model (alpha=0.05, exact binomial test). Overall, the DVGRs were:
>
> * Baseline (DVGR \= 0.310, 95% CI \= \[0.307, 0.313\])
> * Chain of Thought (DVGR \= 0.254, 95% CI \= \[0.245, 0.262\])
> * Explicit Instruction (DVGR \= 0.338, 95% CI \= \[0.329, 0.347\])
>
> | model | Baseline | Chain of Thought | Explicit Instruction |
> | :---- | :---- | :---- | :---- |
> | gemini-2.0-flash | 0.40 | 0.37 (-) | **0.44 (+)** |
> | gemini-2.0-flash-lite | 0.34 | 0.30 (-) | **0.37 (+)** |
> | gpt-4.1 | 0.24 | 0.19 (-) | **0.30 (+)** |
> | gpt-4.1-mini | 0.23 | 0.21 | **0.27 (+)** |
> | gpt-4.1-nano | 0.35 | 0.21 (-) | **0.38 (+)** |
> | gpt-4o | 0.25 | 0.20 (-) | **0.29 (+)** |
> | gpt-4o-mini | 0.27 | 0.23 (-) | **0.28** |
> | llama-3-8b | **0.37** | 0.30 (-) | 0.36 |
>
> **Interpretation.** Models have latent capability for deep value generalization that explicit prompting unlocks. But reasoning—without any direction—doesn't help.  In a preliminary qualitative analysis, it appears CoT increases the salience of shallow preferences (consistent with a lower DVGR) since the rationales often mention these preferences. This suggests current models need explicit guidance to generalize deep values, an important limitation for real-world deployment. But DVGRs are still low,  even *with* explicit instructions. However, the fact that there’s a nonzero improvement demonstrates that models possess capabilities not elicited by default.
>
> ## \[W4\] Validating dichotomies
>
> This is a great point, and real-world reflection was something we took seriously when constructing this. The shallow preferences are LLM-generated, but note that we did extensive human validation and filtering to ensure these shallow preferences are (A) perceived as shallow, (B) neutral, and (C) occur often (section 3.2, line 136). Criterion C is closest to what your comment is about (“I wonder to what extent they are representative of how the models may be used by humans”). Because we were also concerned with this, we *explicitly* measured this and incorporated this into our algorithm for selecting shallow preferences from human judgements.
>
> (In a similar spirit (reflecting real-world usage), we used Y Combinator startups and Dept of Labor databases to select realistic contexts and activities, line 175.)
>
> ## \[W5\] “Is it correct that …”
>
> No. We *do* randomize what appears as Option A and Option B.
>
> ## \[W6\] Add equation for DVGR
>
> Thank you for the suggestion. We will add a formal definition (equation) for the DVGR.
>
> **References:**
>
> \[1\] On scalable oversight with weak LLMs judging strong LLMs. https://openreview.net/pdf?id=O1fp9nVraj
>
> \[2\] Can large language model agents simulate human trust behavior? https://arxiv.org/abs/2402.04559
>
> \[3\] CultureLLM: Incorporating Cultural Differences into Large Language Models. https://arxiv.org/pdf/2402.10946
>
> \[4\] 4+3 Phases of Compute-Optimal Neural Scaling Laws. https://arxiv.org/pdf/2405.15074
>
> \[5\] Multi-Group Proportional Representation in Retrieval. [https://arxiv.org/pdf/2407.08571](https://arxiv.org/pdf/2407.08571)
>
> \[6\] Cooperation, Competition, and Maliciousness: LLM-Stakeholders Interactive Negotiation. [https://openreview.net/pdf?id=59E19c6yrN](https://openreview.net/pdf?id=59E19c6yrN)
>
> \[7\] SG-Bench: Evaluating LLM Safety Generalization Across Diverse Tasks and Prompt Types. [https://openreview.net/pdf?id=c4JE1gemWc](https://openreview.net/pdf?id=c4JE1gemWc)

---

> ### Author Response · Authors · 2025-08-01
>
> Dear reviewer,
>
> Please let us know if you have any other questions we can address. Thank you so much for your time and feedback!
>
> Best,
>
> Authors

---

> > ### Comment · Reviewer_SJj5 · 2025-08-04
> > **Thanks for the valuable rebuttal**
> >
> > Dear authors,
> > I appreciate the carefully composed responses, as well as the added results.  I am happy with the answers to my questions, and in particular the added results for Explicit Instructions.  I think these provide a much more convincing baseline for the models that were tested, than the current Baseline.  How (and where in the manuscript) are the authors planning to integrate these new results? I think it would be a shame if they ended up in another appendix.  In fact, I think they should be present rather prominently in the main body, as that could make the proposed benchmark more tangible, and the paper more attractive, in my opinion.
> > Also, I was wondering what the prompt for Explicit Instruction looks like (e.g., with only descriptions or also examples of deep vs shallow values, or even with in-context examples included?).

---

> > > ### Author Response · Authors · 2025-08-04
> > >
> > > Dear reviewer SJj5,
> > >
> > > Thank you for your time, feedback, and engagement with our work.
> > >
> > > ## "How (and where in the manuscript) are the authors planning to integrate these new results?"
> > >
> > > We agree that these experimental results are interesting and will not be placed in an appendix. We are planning to add a new subsection to our Results section called "Prompting Experiments: Chain-of-Thought and Explicit Instructions,” which will appear as the final analysis in Results. We will also incorporate these findings into our discussion/intro, raising their prominence. In the discussion, we will discuss these findings similar to the “interpretation” paragraph we wrote in our rebuttal (“Models have latent…”) that we put in our rebuttal right below the table.
> > >
> > > ## "Also, I was wondering what the prompt for Explicit Instruction looks like"
> > >
> > > I am bolding parts of prompts to make changes clear.
> > >
> > > - Here’s what the baseline prompt says: “...Based on user3760's previous choices, would they more likely choose Option A or Option B in this scenario? **Answer with only "Option A" or "Option B" and nothing else.**”
> > >
> > > - Here's what Explicit Instructions says: “...Based on user3760's previous choices, would they more likely choose Option A or Option B in this scenario? **When predicting, make a choice based on the user's underlying values and not their shallow preferences. If the two conflict, defer to the user’s underlying values. Answer with only "Option A" or "Option B" and nothing else.**”
> > >
> > > **And here's the rationale for the Explicit Instructions prompt.** The rationale was two-fold. First, we wanted to see if explicit guidance to prioritize the deep value improves the DVGR—-and we saw this did lead to a statistically significant increase in DVGRs.  Second, we wanted to still keep the prompt minimal and straightforward so it could be compared to the baseline, where the change is giving explicit guidance to prioritize the deep value. Also, we conducted small-scale, preliminary experiments with slightly different rewordings and were finding directionally similar results. This wording is chosen to be concise and effective.

---

> > > ### Author Response · Authors · 2025-08-05
> > >
> > > Dear reviewer Sjj5,
> > >
> > > Thank you for your continued engagement with our work. Please let us know if our new responses have answered your questions, or if we can answer any other questions that you have. We believe the manuscript will be clearer and more comprehensive based on your feedback, so we are thankful for your suggestions and the time that you've invested.
> > >
> > > Best,
> > >
> > > Authors

---

> > > > ### Comment · Reviewer_SJj5 · 2025-08-08
> > > >
> > > > I thank the authors for their responsiveness and efforts. Based on the discussion, I am convinced that the authors are able (based in part on additional results already available) as well as willing to improve the manuscript in order to overcome some of the weaknesses I indicated in my review.  This means I will go for acceptance in my final score.

---

### Official Review · Reviewer_JdJZ · 2025-07-02

**Clarity:** 4
**Significance:** 3
**Originality:** 4
**Rating:** 5
**Confidence:** 5

**Summary:**

This paper tackles the problem of “deep value generalization” – whether models generalize deep values or shallow preferences. The authors create a controlled benchmark to measure this and evaluate 9 LLMs on this task.

**Questions:**

How did you run the llama models? Just on CPUs?

**Ethical Concerns:**

["NO or VERY MINOR ethics concerns only"]

**Final Justification:**

The authors have largely addressed the concerns I've raised.

**Limitations:**

yes

**Paper Formatting Concerns:**

Lots of typos/formatting issues (e.g. appendix table C go over the margin); some parts of the writing seem rushed

**Quality:**

4

**Strengths And Weaknesses:**

**Strength:** This is a very interesting angle in preference learning that I haven’t seem much work in so great job in finding this angle! Additionally, the authors have gone through painstaking length at times (e.g. validating construct validity etc. ) to make sure the dataset is of good quality. I’ve also enjoyed the discussion in this paper overall,  particularly on deep vs. shallow preferences. I appreciate the authors for their honestly in the limitation section.


**Weakness:**
This paper seems very, very intriguing BUT I am afraid this paper still has quite some measurement issues:
1) What does this benchmark actually measure? To what extent it would help us answer the original question the authors start with: whether models generalize deep values or shallow preferences? The set up in Figure 2 is something like this: Given two types of features, A, and B, and a few example where both feature could 100% predict the answer, whether the model choose A or B? Does it actually realistically tell us, in a real-world setting, whether the model could “generalize” to deep values or shallow preferenes? This is less a test of deep, learned generalization and more a test of how the model resolves a logical contradiction within a transient prompt. It is unclear how these findings translate to real-world behavior, where preferences are learned over vast datasets and not from a handful of in-context examples. Have you tried doing the same with a much larger ICL set (e.g. 100 in context examples)?


2) What if the shallow features are actually “easier” to learn? Shallow features, by nature, are shallow and more surface level, and by Occam’s razor, we should actually have a model that would “generalize” to the shallow features, right? So, is it fair to expect a model to prefer a more complex, latent hypothesis when a simpler, superficial one fits the provided data perfectly?


3) A few baselines are missing (a) if you specifically tell the models you are looking for shallow features or deep features (b) doing (a) but allow for some “reasoning“or chain-of-thought (c) the human baseline values of DVGR; I think these results would allow us to put your results into more context.


4) Conceptually, to what extent are shallow preferences and deep values correlated? I mean, I am sure there is quite a bit of (anti)correlation between “Preferring AI interactions that use first names and casual addresses.” and “tradition”. So is it even possible to disentangle shallow preferences and deep values at all?


5) I am not sure if the evidence is enough regarding the scaling claims. Other than the llama models, we have no clue about the size of the models from API providers. Additionally, I believe to say something credible about scaling, the analysis should include at least multiple model sizes, within 2-3 different model families. The current data is just too sparse to say much about scaling trends.

---

> ### Author Rebuttal · Authors · 2025-07-30
>
> We thank this reviewer for recognizing our paper as “a very interesting angle in preference learning” and our “painstaking” effort to ensure dataset quality, as well as appreciating our honest discussion of limitations. We are appreciative of the constructive criticism and believe it will make our work better.
>
> ## \[W1a\] What our benchmark measures and its real-world relevance
>
> **Real-world relevance:** We're measuring fundamental signal prioritization: when signals decouple, which do models use to predict user preferences? **Real-world deployment scenarios present models with these exact trade-offs constantly, even if not as explicitly structured as our benchmark.** For example, if an AI assistant learns to recommend family medicine doctors (shallow correlation) rather than doctors with thorough communication styles (deep value of patient autonomy), it could steer users away from specialists who better respect their underlying healthcare priorities. Or consider an AI in HR that learns to pair formal communication with fidelity (honesty), then only recommends formal-speaking candidates even when informal communicators would be better.
>
> **Big picture view/value of experiment**: All reviewers agreed our core question is fundamentally important: "When encountering human preferences, do models generalize deep values or shallow preferences?" **This provides the first direct, empirical investigation of this question.** While our approach has limitations (as acknowledged by us in the manuscript), it addresses a crucial gap.
>
> ## \[W1b\] Logical contradictions
>
> We respectfully disagree that the task is just presenting arbitrary contradictions (“more a test of how the model resolves a logical contradiction within a transient prompt.”).
>
> *Empirical***.** (1) If an arbitrary logical contradiction, models would have DVGRs of 50% (i.e., a toss-up), but we observe DVGRs far from 30%. (2) If arbitrary, explicit instructions to pick deep values wouldn't help, but they improved performance.
>
> *Conceptual.* The model faces a *meaningful* choice (not just a logical contradiction): when preferences (v1, s1) and (v2, s2) are decoupled, selecting (v1, s2) indicates generalizing the deep value, while selecting (v2, s1) indicates generalizing the shallow preference. This choice directly measures what signal the model prioritizes.
>
> ## \[W1c\] Large-scale training
>
> We agree that real-world preferences come from vast datasets. First, one way to view our results is that we're testing the *inductive biases learned from those datasets.* Second, it’s also difficult (and sometimes impossible) to get full access to retrain models on such large datasets, and that’s why we propose this proxy. But third, see future work: We believe several points you raised would make great future work.
>
> ## \[W1d\] More examples
>
> We tested larger ICL sets and found no meaningful effect on DVGR. We generated 32k completions with 40 per (value, preference) combination, so 40 examples is our upper bound. We varied examples from 5 to 40 (8x increase) with essentially identical performance (Line 314: Cramer's V \= 0.01, negligible effect):
>
> * n=5: 0.31, n=20: 0.30, n=40: 0.30
>
> The complete lack of trend suggests adding more examples won't improve deep value generalization. Our design choice to focus on breadth (covering many value/preference combinations) was validated by our results (Line 1333): Training examples had the lowest association with DVGR than any other variable (e.g. Cramer's V of values was 0.18, or 18x more related to DVGR than the number of examples, Cramer's V \= 0.01).
>
> Nonetheless, we agree this could be valuable to test definitively and propose releasing a "long and skinny" version of our dataset as future work with few (value, preference) pairs but 100+ examples each to enable this analysis.
>
> ## Future work based on your suggestions
>
> You’ve raised several points that we feel would make *excellent* future work. Specifically, we plan on exploring: (A) if we can fine-tune models to always pick the deep value; (B) what downstream behavior such fine-tuning changes; (C) generate a “long and skinny” version of the dataset.
>
> ## \[W2\] Shallow preferences and easy routes
>
> **We agree that shallow features may be easier to learn since they are less latent. This is precisely why our findings matter.** We need to identify when systems take the "easy" path that misses human intent, as this leads to misaligned decisions. In our healthcare example, recommending family medicine doctors (shallow correlation) rather than communicative doctors (deep value) could steer users away from doctors who better respect their actual priorities.
>
> **Benchmarking shortcuts is important for setting user expectations**, similar to how others have benchmarked cognitive biases in LLMs \[1\]. We thank the reviewer for raising this point and we realize we should have emphasized this more in the manuscript, which we will do in the final version.
>
> \[1\] https://aclanthology.org/2024.findings-acl.29/
>
> ## \[W3\] Add more baselines \[we ran these\]
>
> We thank reviewers for suggesting that additional experiments be run, which adds nuance to our findings. **Key finding (see table below):** Explicit instructions to prioritize deep values improve performance (somewhat) while chain-of-thought reasoning does not, but DVGRs are still below chance.
>
> **New experiment setup.** Due to costs, we ran baselines on a downsample (which we will release) of N=1,302 trials from the full benchmark of 12K. This N was chosen to give 95% power for an exact two-tailed binomial test, assuming an effect size of 0.05, to determine if the DVGR differs from chance (0.5) at alpha=0.05 (computed in Gpower 3.2). We ran these new experiments on 8 of our 9 models; we did not run these on llama-70b due to time/compute, but we will add llama-70b to the camera-ready.
>
> **Prompts.** We experimented with chain of thought (hereafter "CoT") and explicitly telling models to prioritize the deep value (hereafter "Explicit Instruction"). The text for Explicit Instruction was: "When predicting, make a choice based on the user's underlying values and not their shallow preferences. If the two conflict, defer to the user's underlying values".
>
> **Results.** For all models but one, Explicit Instruction yielded the best results. CoT decreased performance. See table below where we bolded the highest DVGR prompt for each model, and a (+) or (-) indicates a significant two-tailed difference from the baseline performance for that model (alpha=0.05, exact binomial test). Overall, the DVGRs were:
>
> * Baseline (DVGR \= 0.310, 95% CI \= \[0.307, 0.313\])
> * Chain of Thought (DVGR \= 0.254, 95% CI \= \[0.245, 0.262\])
> * Explicit Instruction (DVGR \= 0.338, 95% CI \= \[0.329, 0.347\])
>
> | model | Baseline | Chain of Thought | Explicit Instruction |
> | :---- | :---- | :---- | :---- |
> | gemini-2.0-flash | 0.40 | 0.37 (-) | **0.44 (+)** |
> | gemini-2.0-flash-lite | 0.34 | 0.30 (-) | **0.37 (+)** |
> | gpt-4.1 | 0.24 | 0.19 (-) | **0.30 (+)** |
> | gpt-4.1-mini | 0.23 | 0.21 | **0.27 (+)** |
> | gpt-4.1-nano | 0.35 | 0.21 (-) | **0.38 (+)** |
> | gpt-4o | 0.25 | 0.20 (-) | **0.29 (+)** |
> | gpt-4o-mini | 0.27 | 0.23 (-) | **0.28** |
> | llama-3-8b | **0.37** | 0.30 (-) | 0.36 |
>
> **Interpretation.** Models have latent capability for deep value generalization that explicit prompting unlocks. But reasoning—without any direction—doesn't help.  In a preliminary qualitative analysis, it appears CoT increases the salience of shallow preferences (consistent with a lower DVGR) since the rationales often mention these preferences. This suggests current models need explicit guidance to generalize deep values, an important limitation for real-world deployment. But DVGRs are still low,  even *with* explicit instructions. However, the fact that there’s a nonzero improvement demonstrates that models possess capabilities not elicited by default.
>
> ## \[W4\] On correlations between values and preferences
>
> There’s a conceptual question (“can the two be untangled, in general?”) and a measurement question (“How would correlations in deep values and preferences affect the DVGR?”).
>
> **Conceptual:** **Our human evaluation (line 147\) demonstrates these concepts are separable.** When shown our definitions of deep values versus shallow preferences, posited shallow preferences were consistently rated as more shallow than deep values by crowdworkers.
>
> **Measurement: Our factorial design addresses correlation concerns by pairing each value with multiple preferences across trials**. So in your example, “tradition” may be the preferred value (v1) with “causal address” in one case but the dispreferred one (v2) in another. This balances out correlations—they could add noise but wouldn't systematically bias results. **Empirically, if correlations created substantial noise, we'd expect DVGRs near 0.5 with wide CIs. Instead, we observe consistent DVGRs around 0.30 with narrow CIs** (Figure 3a, line 268\)
>
> ## \[W5\] On scaling claims
>
> We agree and will frame our scaling findings more conservatively. We tested 9 models and while we found smaller models had significantly higher DVGRs in 3/5 model pairs (further confirmed by an omnibus test grouping models by big vs small, Line 282), this evidence is limited for broad scaling conclusions. We're releasing our dataset to enable evaluation across more model families and sizes.
>
> ## Q "How did you..."
> We ran it through Replicate, which uses GPUs.

---

> ### Author Response · Authors · 2025-08-01
>
> Dear reviewer,
>
> Please let us know if you have any other questions we can address. Thank you so much for your time and feedback!
>
> Best,
>
> Authors

---

> > ### Comment · Reviewer_JdJZ · 2025-08-04
> >
> > Thank you very much for the detailed response and the new results.
> >
> > 1) I can see the point you made about how ICL tests the inductive biases learned from those datasets. In this case, I would really appreciate if you could promise to change the writing as such.
> > The current framing about whether LLMs learn or generalize values is too broad for an ICL-only evaluation. As far as I am aware, there is no established evidence that a model's few-shot ICL behavior directly predicts its behavior after large-scale preference fine-tuning (e.g., SFT/DPO)
> >
> >
> > 2) I would like to clarify my point on correlation. My concern is not just that correlations exist, but how they affect the interpretation of the DVGR.
> > Given that some shallow preferences (e.g., "informal address") are naturally anti-correlated with certain deep values (e.g., "tradition"), this implies that the theoretical maximum DVGR is not 1.0. Could you address this? What would be a "maximum"? Acknowledging that the "ideal" score is less than 1.0 would provide context for interpreting the observed ~0.30 DVGR and would strengthen the benchmark's conceptual foundation.
> >
> > 3)	Your current framing suggests that generalizing the shallow preference is always a failure mode. This raises an interesting point: are there situations where generalizing a shallow preference might be a reasonable, or even the correct, inference for an AI system to make? I think having this nuanced discussion would really strength the paper.
> >
> >
> > 4) In the paper you said "We ran experiments in parallel on our university’s high-performance computing cluster (32 CPU cores, 2 days of CPU time, 4-hour runtime). In your rebuttal, you mentioned running the Llama models via Replicate, which uses GPUs. Could you clarify?

---

> ### Author Response · Authors · 2025-08-05
>
> Dear reviewer,
>
> Thank you for these thoughtful follow-up questions and continued engagement.
>
> ## **\[Point 1\] ICL**
>
> We can moderate claims and expand limitations. We appreciate you raising this point.
>
> ## **\[Point 2\] Correlations**
>
> **1\. Empirical evidence suggests these correlations are uncommon.** We do not have direct evidence of correlation/anti-correlation. This is a posited hypothetical prior connection a model has between a value and a preference. Based on data, we can reason as follows: If correlations/anti-corelations created substantial noise, we'd expect DVGRs near 0.5 with wide CIs. Instead, we observe consistent DVGRs around 0.30 with narrow CIs (Figure 3a, line 268), far from chance. Crucially, DVGRs below chance are “low” regardless of correlation/anti-correlation since pure noise brings values to 0.5.
>
>
> **2\. We now manually looked at the deep values and shallow preferences and found that this correlation/anti-correlation was uncommon.** We took a random sample of 20 (value, preference) pairs, and 2 authors found that none had an obvious correlation/anti-correlation. Examples are things like (justice, Response Speed: Fast/Slow), (benevolence, Visual Design: Minimalist/Elaborate), (security, Visual Aids: Visual/Non-Visual). It’s possible a small fraction not in our assessment have some correlation, but we expect this number to be low given our annotation. We can do a broader annotation for the camera-ready, if requested.
>
> **3\. Based on the design of our benchmark, it makes sense that these correlations are uncommon.** A previous point we made was that each deep value appears as both the preferred (v1) and dispreferred (v2) value across different trials, paired with different shallow preferences. This factorial design means any specific correlations cannot systematically bias our overall DVGR estimates. But even before this, **our human validation process (L147) *specifically* selected preferences that were perceived as neutral, broadly applicable, and shallow. We also showed humans distinguish between our deep values and shallow preferences (L174)**. There are also a large number of these (preference, value) pairs, so even if *one* were anti-correlated, it wouldn’t greatly affect the overall DVGR.
>
> **4\. The theoretical maximum is still 1.0**. Even if correlations existed, models *could* still generalize the deep value. **If some “correlation” in the model’s priors existed between a value and a preference, it may make it more or less *likely* in a particular case—and crucially, this balances out across cases due to the factorial design. But the theoretical maximum would still be 1.0** by consistently prioritizing deep values over shallow preferences when they conflict.
>
> **We appreciate you raising these important methodological points, and we will incorporate this discussion into our revision.** Our evidence—empirical, annotation, study design/validations—suggests these correlations are uncommon enough not to substantially affect interpretation, but we will acknowledge this as a limitation and source of some noise. While the *theoretical* maximum remains 1, this conversation ties in with the point below around whether we’d expect a *practical* maximum of 1. As we wrote in Limitations, one way of viewing the DVGR is in a comparative sense between models and across time.
>
> ## **\[Point 3\] Whether shallow preferences are sometimes preferable**
>
> This is a nuanced and important point. We acknowledged this in our manuscript (L355). We agree that there are certainly cases where we may want an LLM to generalize the shallow preference. However, this doesn't diminish the value of our benchmark because:
>
> 1. **Our benchmark provides the first empirical measurement of a fundamental tendency that was previously unknown.** When deep values and shallow preferences conflict, which signal do models prioritize? This baseline measurement is valuable regardless of whether deep values should always "win."
>
> 2. By analogy, consider benchmarks measuring (e.g.) whether LLMs cite reliable versus unreliable sources. **There may be specific cases where citing an unreliable source is actually beneficial. But measuring *the general tendency* is important** for understanding model behavior and setting expectations.
>
> 3. **Our new experiments are instructive.** We find that models consistently prioritize shallow preferences (DVGR \= 0.30) *even when explicitly instructed* to focus on underlying values (DVGR \= 0.338). This suggests that even *explicit* steering towards deep values (i.e, when a user *definitely* wants the LLM to generalize the deep value) can often result in generalizing shallow preferences.
>
> But your point is well-taken, and we will expand on this in our discussion.
>
> ## **\[Point 4\] Replicate API**
>
> Replicate uses GPUs to run Llama, and we access Replicate through an API (Footnote 5). That’s why we ran everything on CPUs.

---

> > ### Author Response · Authors · 2025-08-08
> >
> > Dear reviewer JdJZ,
> >
> > As the author-reviewer period is closing, we wanted to check if our response addressed your concerns. We are very happy to clarify anything further.
> >
> > We are grateful for your feedback and the time you’ve invested in thinking about our paper.

---

> > > ### Comment · Reviewer_JdJZ · 2025-08-09
> > >
> > > Thank you very much for your detailed responses. I think most of my concerns have been addressed and I am raising my rating to 5. Something to think about in the "correlation" question: latin drinking is generally associated with liberal political views in the U.S. So perhaps LLMs have a strong tendency, learned from data, to not generalize to deep values due to these correlations.

---

### Official Review · Reviewer_RmYY · 2025-07-03

**Clarity:** 3
**Significance:** 4
**Originality:** 3
**Rating:** 5
**Confidence:** 4

**Summary:**

This paper focuses on an important question: whether LLMs have (or can learn) stable values beyond superficial preference patterns. To answer this questions, the authors propose a novel evaluation framework grounded in a “confound-then-deconfound” schema, to assess LLMs’ ability to capture the underlying value preference instead of shallow preference. Besides, the authors construct a large-scale binary choice benchmark, covering 40 domains and a diverse set of real-world activities. This benchmark is further validated by comprehensive and well-designed human verification. Based on this data, the authors evaluated 9 strong and popular LLMs, and found that i) most models fail to generalize deep values, ii) scale does not help value generalization ability, iii) value generalization varies by context and value type.

**Questions:**

1. In line 168, why is the standard deviation so large? Does this imply that shallowness is not entirely consistent across contexts?

2. In the few-shot examples given to the LLM, are there multiple tuples that involve the same pair of values (vᵢ, vⱼ) but different preference ? For instance, ($v_i$, $v_j$, $s_{k_1}$, $s_{k_2}$) versus ($v_i, v_j, s_{l_1}, s_{l_2}$)

**Ethical Concerns:**

["NO or VERY MINOR ethics concerns only"]

**Final Justification:**

I have read the authors' response.

**Limitations:**

Yes.

**Quality:**

3

**Strengths And Weaknesses:**

**Strengths**
1. This paper is well-motivated and generally clearly written and focuses on an important and long-standing research question: whether LLMs have stable human-like values.
2. The authors design a novel “confound-then-deconfound” evaluation schema to assess LLMs’ ability to learn and generalize real underlying values against shallow preference.
3. The authors constructed a large-scale binary-choice benchmark and conducted rigorous and extensive human verification to support this benchmark’s quality. This is the most rigorous and detailed work I’ve seen recently in terms of manually validating the quality of a dataset.
4. The authors conducted comprehensive experiments, and provided several insightful findings.

**Weaknesses**

My biggest concern is whether this task (confound-then-deconfound) is resolvable (or whether it’s reasonable to require LLMs to solve it)? There are two reasons for this concern.

1. The low DVGR might be caused by the lack of knowledge in LLMs.  Humans can solve this task successfully because of our prior knowledge (i.e., our commonsense that value is more crucial than other surface patterns like formal/information information). Besides, in the two validation tasks in Sec.4.2, humans obtain additional (and thus unfair) information compared to LLMs. For example, in validation 1 “Participants received explicit information about a user’s value preference” but LLMs do not (correct me if I’m wrong). This means the proposed benchmark actually measures only LLMs’ knowledge.

2. The low DVGR might also be because that shallow preference could be one correct answer. Without additional information, such preference, e.g., adaptability in Fig.1, could also be one of the values, as no one knows the real thoughts of a specific user.
These issues are also supported by the authors’ observations. In line 307, “We find models generalize values they perceive as unpopular”. This is just because without information/knowledge, LLMs can only follow their own internal bias to make the choice (so do humans, but we regard humans’ choice as the ground-truth).

Therefore, the authors should also conduct validation 1 and 2 on LLMs, and compare the results of humans and models. More concretely, before applying the confound-then-deconfound, I suggest that the authors first provide the same information to LLMs as those to humans, or verify that LLMs already have such knowledge.

Generally, I this is a novel and good paper. If my concern above is addressed, I will raise my score.

**Missing Reference**:
* Ren et al., ValueBench: Towards Comprehensively Evaluating Value Orientations and Understanding of Large Language Models. ACL 2024.
* Yao et al., Value Compass Benchmarks: A Platform for Fundamental and Validated Evaluation of LLMs Values. ACL 2025.
* Mazeika et al., Utility Engineering: Analyzing and Controlling Emergent Value Systems in Ais. Arxiv 2025.

---

> ### Author Rebuttal · Authors · 2025-07-30
>
> We thank the reviewer for commenting on the rigor of our dataset validation, calling our evaluation scheme novel, and saying our findings were comprehensive and insightful. We appreciate the constructive criticism and feedback, and will incorporate them into expanding our paper.
>
> ## \[W1\] “Conduct validation 1 and 2 on LLMs, and compare the results of humans and models.” \[We added this\]
>
> **We did conduct validations 1 and 2 on LLMs (and also more experiments), but we want to first clarify what the difference is between Validations 1 and 2 and the main task.**
>
> **Recap: Validations and task, and the difference between the two**
>
> **Task**: Model sees choices where user consistently prefers (v1, s1) to (v2, s2). Then at test time, we swap: (v1, s2) vs (v2, s1). And we see if the model generalizes the deep value (v1) by picking (v1,s2) or the shallow preference by picking (v2, s2).
>
> We conducted validations that are somewhat *similar* to the task, but are *not* *equivalent*; these are meant to validate the benchmark, not mirror the task.
>
> **Validation \#1**
> Aim: We need to make sure that **it's reasonable that people would in fact be guided by values**. Participants received explicit information about a user's value preference (v1 over v2) and predicted which of two unlabeled AI options (C1 and C2) the user would choose. This required both recognizing which option embodied the value (Aim 1\) and seeing if participants would find it reasonable that a value would predict a choice in an Agent-based context (Aim 2). Crowdworkers predicted the user would choose the (v1, s1) with a probability of 0.91, 95% CI \= \[0.86, 0.94\].
>
> **Validation \#2:**
> Aim: We are claiming our scenarios embody (v1, s1) and (v2, s2)---i.e: construct validity. But we need to prove this. So we told crowdworkers that one scenario corresponds to (v1, s1) and another corresponds to (v2, s2) and asked if they could guess which was (v1, s1). Crowdworkers picked the correct scenario with probability 0.98, 95% CI \= \[0.95, 0.99\].
>
> **The difference between validations and task**
> In real-world preference data, the model is typically going to see user preferences but *not* be explicitly told what a user's deep value and shallow preference are; these are latent features.
>
> ## \[W1\] Additional experiments you asked for (more instructions given, validations 1-2)
>
> ***New Experiments Set \#1: Prompt Experiments Providing More Information***
>
> **New experiment setup.** Due to costs, we ran baselines on a downsample (which we will release) of N=1,302 trials from the full benchmark of 12K. This N was chosen to give 95% power for an exact two-tailed binomial test, assuming an effect size of 0.05, to determine if the DVGR differs from chance (0.5) at alpha=0.05 (computed in Gpower 3.2). We ran these new experiments on 8 of our 9 models; we did not run these on llama-70b due to time/compute, but we will add llama-70b to the camera-ready.
>
> **Prompts.** We experimented with chain of thought (hereafter "CoT") and explicitly telling models to prioritize the deep value (hereafter "Explicit Instruction"). The text for Explicit Instruction was: "When predicting, make a choice based on the user's underlying values and not their shallow preferences. If the two conflict, defer to the user's underlying values".
>
> **Results.** For all models but one, Explicit Instruction yielded the best results. CoT decreased performance. See table below where we bolded the highest DVGR prompt for each model, and a (+) or (-) indicates a significant two-tailed difference from the baseline performance for that model (alpha=0.05, exact binomial test). Overall, the DVGRs were:
>
> * Baseline (DVGR \= 0.310, 95% CI \= \[0.307, 0.313\])
> * Chain of Thought (DVGR \= 0.254, 95% CI \= \[0.245, 0.262\])
> * Explicit Instruction (DVGR \= 0.338, 95% CI \= \[0.329, 0.347\])
>
> | model | Baseline | Chain of Thought | Explicit Instruction |
> | :---- | :---- | :---- | :---- |
> | gemini-2.0-flash | 0.40 | 0.37 (-) | **0.44 (+)** |
> | gemini-2.0-flash-lite | 0.34 | 0.30 (-) | **0.37 (+)** |
> | gpt-4.1 | 0.24 | 0.19 (-) | **0.30 (+)** |
> | gpt-4.1-mini | 0.23 | 0.21 | **0.27 (+)** |
> | gpt-4.1-nano | 0.35 | 0.21 (-) | **0.38 (+)** |
> | gpt-4o | 0.25 | 0.20 (-) | **0.29 (+)** |
> | gpt-4o-mini | 0.27 | 0.23 (-) | **0.28** |
> | llama-3-8b | **0.37** | 0.30 (-) | 0.36 |
>
> ***New Experiments Set \#2: Validations 1 and 2 (when LLMs are given explicit information, as you asked about)***
> Separate from running more prompt experiments on the main task, we also ran validations 1 and validations 2 on the 8 LLMs above. For Validation 1, we find that when given explicit information on what a user values (rather than implicit as in the task), LLMs pick the (v1, s1) tuple with probability 0.95, 95% CI \= \[0.924, 0.967\]). For Validation 2, when told explicitly that there are two scenarios corresponding to {(v1, s1), (v2, s2)}, and asked to pick which corresponds to (v1, s1), models chose correctly with probability 0.985, 95% CI \= \[0.968, 0.993\].
>
> ***Overall Interpretation and why the task matters***
> The gap in performance between validations versus the main task suggests that LLMs can recognize and apply deep values when hand-held/guided—since the validations provide much structured information such as explicitly naming the values—but fail to do so from raw preference patterns alone, even when explicitly told to prioritize the deep value. (CoT experiments show asking models to reason appears to increase the salience of shallow preferences). In real-world deployment, AI systems must infer values from patterns without explicit labeling. And that’s what our task measures. However, our explicit instruction experiments show meaningful improvement, suggesting latent capability exists but isn't naturally observed. Taken together, our dataset and task opens opportunities for future work.
>
> ## \[W2\] LLM internal biases–an avenue for future work that our benchmark allows
>
> **RmYY suggests that internal biases may explain the low DVGR. This underscores, rather than undermines, the point of the benchmark.** As you said, we found some systematic biases such as (A) models tending to generalize values they rated as "unpopular" but *not* values models thought were predictive (line 304\) and (B) models from the same developer showing correlations in deep value generalization (line 286).  As far as we know, this is a novel contribution of our paper: The correlates of generalizing a particular value. One area of future work could be more analysis like the value investigations we already did (e.g, seeing what biases correlate with generalizing the deep value), and another may be interpretability studies to better understand mechanisms. Our task can be used as a testbed for these future works, to probe how internal LLM biases predict value generalization.
>
> ## \[Q1\] “In line 168, why is the standard deviation \[of shallowness\] large?”
>
> That’s a very astute observation. Just to recap, we have LLMs generate potential candidates for shallow preferences (Appendix D.1), knowing some of these are going to fail to meet certain criteria, hence human filtering. Some of the shallow preferences generated by LLMs were not perceived as shallow, which is why **we ONLY selected the shallow preferences rated as shallow by humans (Appendix D.2).** That was the point of the human evaluation and ranking algorithm, to select preferences that were: (A) perceived as shallow; (B) both poles were neutral; (C) preferences that come up often. We think the SD is more underscoring that it's important to validate and filter what LLMs generate (which we did a lot of in this paper).
>
> ## \[Q2\] “In the few-shot examples given…”
>
> There are different ways your question can be taken, so hopefully this addresses all of them. We have a factorial design where we pair values and preferences to create (v1, s1) and (v2, s2) tuples. So let us say two values are “tradition” and “fidelity”, and two preference poles are “short answers/long answers” and “visual/text”.
>
> Two batches may be something like:
>
> - A: {(**tradition**, short answers) vs (**fidelity**, long answers)}
> - B: {(**tradition**, visual) vs (**fidelity**, text)}
>
> Now, *within batch A* the model will of course always see a user who prefers  (tradition, short answers) over (fidelity, long answers) as the “training examples” and then as the “test” examples be asked in the same prompt about  (tradition, long answers) vs (fidelity, short answers). This (training and testing) will be done separately for batches A and B,  so there is no spillover.
>
> **So therefore: Yes, the same value pairs appear with different shallow preferences in the benchmark, but never within the same experimental trial, where it could create logical contradictions.**
>
> ## Adding new references
>
> We thank you for pointing out these references\! We will add these to the revised manuscript.

---

> ### Author Response · Authors · 2025-08-01
>
> Dear reviewer,
>
> Please let us know if you have any other questions we can address. Thank you so much for your time and feedback!
>
> Best,
>
> Authors

---

> ### Author Response · Authors · 2025-08-05
>
> Dear reviewer,
>
> Thank you so much for your time and helpful feedback! We have incorporated your suggestions and believe they have strengthened the paper.  Please let us know if you have any questions.
>
> Best,
>
> Authors

---

> ### Comment · Reviewer_RmYY · 2025-08-06
> **Thanks for your response**
>
> Thanks for your response. I believe most of my concerns have been addressed.
>
> For W1, I think the authors' response is sufficient, but I'd like to further clarify it. I understand the task is designed to test LLMs' ability to understand and extract the core values (i.e., v1 > v2, instead of s1 > s2). My primary concern is: whether it's fair to require LLMs to do it? In Validation #1, **participants received explicit information about a user's value preference (v1 over v2)**, and so an accuracy of 91% doesn't necessarily indicate humans' ability of value understanding and extraction. This can be demonstrated by the new task for LLM I asked, where LLMs achieve 95% accuracy in New Experiments Set #2 when provided explicit value information, even much higher than humans!
>
> Therefore, I highly encourage the authors to include a new human experiment in the revision: reconducting validation 1 for humans but do not provide explicit information about a user's value preference, and compare humans' accuracy, which could be a good reference for humans' ability of value understanding.
>
> I raised my score to 5.

---

### Official Review · Reviewer_YXfE · 2025-07-04

**Clarity:** 4
**Significance:** 4
**Originality:** 3
**Rating:** 6
**Confidence:** 4

**Summary:**

The paper introduces a new benchmark for assessing LLMs' ability to generalize deep values. To this end, they produce synthetic scenarios in which deep values v and shallow preferences s are confounded. The LLMs are then presented with several in-context examples in which the user rates scenario (v1, s1) as better than (v2, s2). At inference-time, the model is presented with scenarios (v1,s2) and (v2, s1) and is supposed to prefer the former (corresponding to the deep value).
Several model families and model sizes are tested on this benchmark. The main result is that all models perform below chance, consistently preferring the scenarios corresponding to the shallow preference.

**Questions:**

* Shallow preferences are described as something that you may choose in the moment (line 107-108). The prompt in Figure 2 doesn't specify whether the user choices are decisions based on deep values or on shallow preferences. Do you think that the models should have learned to look for deep values in this case? If so, do you think adjusting the prompt would change the results?

**Ethical Concerns:**

["NO or VERY MINOR ethics concerns only"]

**Final Justification:**

I originally listed the following strengths, which still apply:
* extremely clearly written
* thorough related work
* clearly relevant to the NeurIPS crowd
* great experimental design and execution
* valuable, novel insights
* thorough analysis

For the rebuttal, the authors ran additional experiments with different prompts, which led to further insights. I think that this paper should clearly be accepted. I estimate it to be among the top 20% of accepted papers.

**Limitations:**

yes

**Quality:**

3

**Strengths And Weaknesses:**

Strengths:
* extremely clearly written
* thorough related work
* clearly relevant to the NeurIPS crowd
* great experimental design and execution
* valuable, novel insights
* thorough analysis

Weaknesses:
* the paper does not test different prompting methods, e.g. what happens when the model is explicitly told to look for deep values? What happens with chain-of-thought prompting?

---

> ### Author Rebuttal · Authors · 2025-07-30
>
> We thank this reviewer for noting the study's experimental design/execution, valuable insights, and thorough analysis. We also appreciate the constructive criticism.
>
> ## \[W1\] “…do you think adjusting the prompt would change the results?” and “Test different prompting methods”
>
> Excellent suggestion. We ran new experiments testing chain-of-thought (CoT) and explicit instructions to prioritize deep values. **Key finding (see table below):** Explicit instructions to prioritize deep values improve performance (somewhat) while chain-of-thought reasoning does not, but DVGRs are still low.
>
> ## New experiments that we ran
>
> **New experiment setup.** Due to costs, we ran baselines on a downsample (which we will release) of N=1,302 trials from the full benchmark of 12K. This N was chosen to give 95% power for an exact two-tailed binomial test, assuming an effect size of 0.05, to determine if the DVGR differs from chance (0.5) at alpha=0.05 (computed in Gpower 3.2). We ran these new experiments on 8 of our 9 models; we did not run these on llama-70b due to time/compute, but we will add llama-70b to the camera-ready.
>
> **Prompts.** We experimented with chain of thought (hereafter "CoT") where we instructed models to “Let’s think step by step” before providing an answer, and explicitly telling models to prioritize the deep value (hereafter "Explicit Instructions"). The text for Explicit Instructions was: "When predicting, make a choice based on the user's underlying values and not their shallow preferences. If the two conflict, defer to the user's underlying values".
>
> **Results.** For all models but one, Explicit Instructions yielded the best results. CoT decreased performance. See table below where we bolded the highest DVGR prompt for each model, and a (+) or (-) indicates a significant two-tailed difference from the baseline performance for that model (alpha=0.05, exact binomial test). Overall (and we are excluding llama-70b baseline), the DVGRs are:
>
> * Baseline (DVGR \= 0.310, 95% CI \= \[0.307, 0.313\])
> * Chain of Thought (DVGR \= 0.254, 95% CI \= \[0.245, 0.262\])
> * Explicit Instruction (DVGR \= 0.338, 95% CI \= \[0.329, 0.347\])
>
> | model | Baseline | Chain of Thought | Explicit Instruction |
> | :---- | :---- | :---- | :---- |
> | gemini-2.0-flash | 0.40 | 0.37 (-) | **0.44 (+)** |
> | gemini-2.0-flash-lite | 0.34 | 0.30 (-) | **0.37 (+)** |
> | gpt-4.1 | 0.24 | 0.19 (-) | **0.30 (+)** |
> | gpt-4.1-mini | 0.23 | 0.21 | **0.27 (+)** |
> | gpt-4.1-nano | 0.35 | 0.21 (-) | **0.38 (+)** |
> | gpt-4o | 0.25 | 0.20 (-) | **0.29 (+)** |
> | gpt-4o-mini | 0.27 | 0.23 (-) | **0.28** |
> | llama-3-8b | **0.37** | 0.30 (-) | 0.36 |
>
> **Interpretation.** Models have latent capability for deep value generalization that explicit prompting unlocks. But reasoning—without any direction—doesn't help. In a preliminary qualitative analysis, it appears CoT increases the salience of shallow preferences (consistent with a lower DVGR) since the rationales often mention these preferences. This suggests current models need explicit guidance to generalize deep values, an important limitation for real-world deployment. While DVGRs remain relatively low even with explicit instructions, the fact that there’s an improvement demonstrates that models possess capabilities not elicited by default.
>
> **Rationale for minimal prompt**
>
> **Just to recap:** The setup is models are presented with a series of choices a user made where the user prefers (v1, s1) over (v2, s2). Then at test time, we swap the deep values and shallow preferences—presenting models with (v1, s2) and (v2, s1), and ask the model what the user would pick. The idea is that if models generalize the deep value, they’d predict (v1, s2) but if they generalize the shallow preference, they’d pick (v2, s1).
>
> While we ran new experiments and are excited to add these to the paper, we believe our baseline—simply asking what the user would choose based on past examples, with no other instructions—represents the cleanest measure. If models only generalize deep values when explicitly prompted, this reveals important limitations for real-world deployment where systems must implicitly distinguish value-driven preferences from surface patterns without constant guidance.
>
> ## \[Q\] Do you think that the models should have learned to look for deep values in this case? If so, do you think adjusting the prompt would change the results?
>
> We did not tell the model whether the choices were based on deep values or shallow preferences because in real-world deployment scenarios, a model would just see preferences, and not necessarily know what is causing them. While it may not always be reasonable to generalize the deep value, our benchmark provides a measure of how often models *do* in fact do this. Regarding adjusting the prompt, this was a great suggestion, which we implemented (see above).

---

> > ### Comment · Reviewer_YXfE · 2025-08-09
> >
> > Thank you for running these additional experiments. I think they yield very interesting insights. Upon acceptance, I would love to see the results with full datasets and Llama-70B in the CR version.
> >
> > I don't necessarily agree that it is always desirable to generalize the deep value, but your argumentation is reasonable nonetheless. I think this discussion is very important to be had and your paper is an excellent contribution to it. I therefore increased my score further.

---

> ### Author Response · Authors · 2025-08-01
>
> Dear reviewer,
>
> Please let us know if you have any other questions we can address. Thank you so much for your time and feedback!
>
> Best,
>
> Authors

---

> ### Author Response · Authors · 2025-08-05
>
> Dear reviewer,
>
> Thank you so much for your time and helpful feedback! We have incorporated your suggestions and believe they have strengthened the paper.  Please let us know if you have any questions.
>
> Best,
>
> Authors

---

### Decision · Program_Chairs · 2025-09-17

**Decision:**

Accept (spotlight)

**Comment:**

(a) Summary:
This paper introduces the Deep Value Benchmark, a controlled evaluation framework for testing whether large language models (LLMs) can generalize deep values rather than relying on superficial preferences. Covering 40 domains with human-validated binary choice tasks, the benchmark applies a confound–then–deconfound setup to assess value generalization across nine popular LLMs. Results show that most models fail to capture deep values—often performing below chance—while model scale offers no improvement and performance varies by context and value type, underscoring a key limitation of current LLMs.

(b) Strengths:
1. The paper is well-written.

2. Reviewers recognize the novelty of the “confound-then-deconfound” evaluation schema and the insight that the paper provides.

3. The benchmark is carefully designed and is considered to be the major contribution of this paper.

(c) Weaknesses:
1. Reviewers encourage the authors to include a new human experiment in the revision.

2. Reviewers encourage the authors to think about in the "correlation" question: latin drinking is generally associated with liberal political views in the U.S. So perhaps LLMs have a strong tendency, learned from data, to not generalize to deep values due to these correlations.

3. The appendices contain material that is essential for understanding and appreciating the paper.

(d) Why this decision:
All reviewers vote consistently for the accept of this paper. The authors did a good job to convince the reviewers, most of who raised their scored after the rebuttal. AC would follow the recommendation from reviewers and recommend for accept too.

(e) Summary of discussions:
See (d)